# Pulse Cycle Dependent Nondipole Effects in Above-Threshold Ionization

**Danish Furekh Dar** [1,2,3,*] and **Stephan Fritzsche** [1,2,3]

1   Helmholtz-Institut Jena, D-07743 Jena, Germany; s.fritzsche@gsi.de
2   GSI Helmholtzzentrum für Schwerionenforschung GmbH, D-64291 Darmstadt, Germany
3   Theoretisch-Physikalisches Institut, Friedrich-Schiller-Universität Jena, D-07743 Jena, Germany
*   Correspondence: danish.dar@uni-jena.de

**Abstract:** In this study, we employ strong field approximation (SFA) to investigate the influence of the number of pulse cycles on above-threshold ionization within the framework of nondipole theory. The SFA enables the analysis of the ionization process under the dominance of the electric field, compared to other factors such as the binding potential of an atom. Nondipole effects, including higher-order multipole fields, can significantly impact ionization dynamics. However, the interaction between nondipole effects and pulse cycles remains unclear. Therefore, we investigate the pulse cycle dependence of ionization and examine peak shifts in Kr and Ar atoms. Our findings have implications for comprehensively understanding the effects of electromagnetic fields on electron behavior. The insights gained from this study provide valuable guidance for future research in strong field ionization.

**Keywords:** strong field approximation; above-threshold ionization; nondipole effects; noble gas; few-cycle pulse

## 1. Introduction

The interaction of atoms and molecules, when subjected to high-intensity laser fields, has recently become an area of significant interest. Such interactions encompass the study of fundamental patterns in both electronic and vibrational processes, as well as several other phenomena, including frustrated double ionization, excitation of Rydberg states, correlated electron emission in multiphoton double ionization, and quantum interference and imaging [1–4]. The complexity on the behavior of electrons brought about by strong field interactions is remarkable. These interactions, between atoms and light, provide a deeper understanding of atomic properties and have opened up new opportunities for applications in areas such as spectroscopy [5] and laser-based technology [6].

When subjected to high-intensity laser fields, atoms exhibit complex behavior, such as above-threshold ionization [7,8], high-harmonic generation [9,10], and non-sequential double ionization [11–13]. The time-dependent Schrödinger equation (TDSE) governs these strong-field ionization processes, and various methods are used to solve it, including numerical, classical, and semi-classical methods. One straightforward method is the strong-field approximation [14–16], which is as a semiclassical method. It provides a simplified explanation of the interaction by assuming a classical description of the electromagnetic field. The Coulomb potential of the parent ion in the electron continuum is ignored, and only the influence of the electric field of the laser field is considered. The strong-field approximation (SFA) is favored over other methods for determining the angular and energy-resolved spectra. The transition amplitude is calculated by combining the direct and re-scattering amplitudes. This approximation has been used to calculate both the above-threshold ionization and the high-harmonic spectra for various laser beams in the near to mid-infrared region [17–20].

A typical setup of an ATI experiment is depicted in Figure 1a. Typically, the target being studied is a gas or a solid-state, which has been prepared in a vacuum chamber. A pulsed laser system is used to generate a high-intensity laser field, which has a peak intensity above the threshold for ionization, usually ranging from $10^{13}$ to $10^{15}$ W/cm$^2$. Subsequently, the target is irradiated by the intense laser field, resulting in the ionization of electrons from the atoms or molecules in the target. Finally, the ejected electrons are collected and analyzed using a suitable detector. In ATI spectra, the detection of maxima of the ionization probability is typically accomplished by positioning a detector perpendicular to the optical axis in the polarization plane, as depicted in Figure 1a. However, the dipole approximation utilized disregards the spatial variation of the laser field, and thus does not consider the Lorentz force acting on the ionizing electrons due to the magnetic component of the laser field. This assumption is valid in short-wavelength and low-intensity, typical below 800 nm and below $10^{13}$ W/cm$^2$ respectively, where the Lorentz force is insignificant. However, in instances of intense laser fields with long wavelengths, the Lorentz force can substantially affect the observed spectra, as seen in Figure 1b, and momentum distributions, as demonstrated in studies by Refs. [21,22]. To accurately determine the peak spectra in these scenarios, the detector must be placed slightly off the polar axis ($\theta_p \pm \delta\theta_p$).

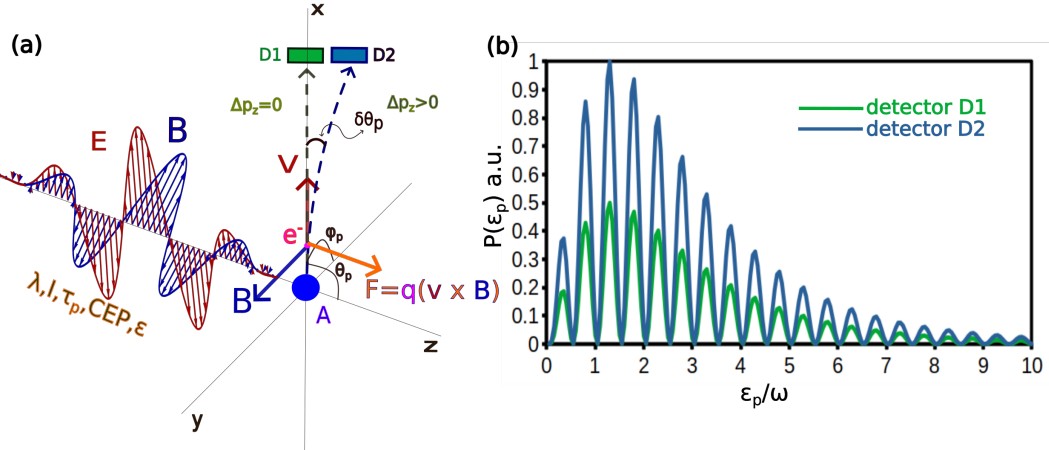

**Figure 1.** A typical setup for ATI measurement considered in our theory with corresponding spectra observed at detector (D1,D2). (**a**) A laser pulse with wavelength ($\lambda$), intensity ($I$), pulse duration ($\tau_p$), carrier envelop phase (CEP), and ellipticity ($\epsilon$) ionizes atom $A$. The ionized electron with velocity ($v$) is then accelerated by an electric field, $E(\mathbf{r}, t)$, with final momentum $\mathbf{p} = (p, \theta_p, \varphi_p)$. The electron experiences a Lorentz force, $F = q(v(t) \times B(\mathbf{r}, t))$, due to a magnetic field, $B(\mathbf{r}, t)$. This causes a shift in the electron's momentum, $\Delta p_z$, and results in detection at a different location, D2, with a change in polar angle $\theta_p$ and corresponding polar shift, $\pm\delta\theta_p$. The azimuthal angle is assumed to be 0. (**b**) The photoelectron energy spectra as observed at D1 and D2.

Recent studies that employ femtosecond pulsed lasers to examine nondipole-induced peak shifts have gained significant attention in the scientific community. This is evident from the numerous references cited in the field [23–27]. These experiments primarily focused on evaluating the shift in the momentum distribution towards the laser propagation. The SFA has traditionally been applied to laser fields under the dipole approximation. However, with the emergence of new research, SFA has been extended to encompass nondipole scenarios, as demonstrated in Ref. [28]. Our recent study [29] further expands this method to include the complicated temporal structure of a few-cycle pulse.

Despite the advancements made, it is crucial to comprehend the impact of pulse duration on the observed nondipole induced peak shifts. The dependence of nondipole induced peak shift on pulse length in ATI has practical implications. The experiments, also mentioned earlier, often employ ultrashort pulses for precision control over strong-field interactions. By uncovering the mechanisms responsible for the pulse length dependence, we can optimize laser parameters to achieve desired experimental outcomes, such as con-

trolling the energy distribution of photoelectrons or enhancing specific ionization processes. Therefore, this study aims to investigate the effect of pulse duration on the observed peak shifts in photoelectron momentum for Krypton and Argon. Understanding the nondipole-induced shifts is essential to gain a comprehensive understanding of photoelectron spectroscopy and to interpret the experimental results. To achieve the objective, we varied the pulse duration and observed the corresponding changes in the peak shifts. Our findings provide valuable insight into the relationship between pulse duration and peak shifts.

It is important to note that this paper uses atomic units, with $m_e = \hbar = \frac{e^2}{4\pi\epsilon_0} = 1$, unless otherwise stated.

## 2. Theory

### 2.1. Introduction to Strong Field Approximation

In this section, we focus on the theory of laser–atom interactions. Our approach combines classical treatment for the laser field with quantum mechanics for the atom. By treating the laser field classically and the atom quantum mechanically, we can obtain a valuable approximation, particularly in the case of intense laser fields where the number of photons per laser mode is high. When an atom is exposed to a laser field, the electric field of the laser can ionize the atom. The ionized electron experiences acceleration due to the electric field of the laser. The acceleration phase of the ionized electron is characterized by its motion in the laser field, which can be described by classical or quantum mechanical models. In addition, if the energy of the photons is not sufficient to ionize the atom, with increasing intensity, multiple photons can be absorbed to overcome the ionization threshold, resulting in multiphoton ionization. However, in ATI, multi-photon absorption goes beyond the ionization potential by more than one photon ($\hbar\omega$), in addition to accessing the ionization continuum. In theoretical framework, such a process is governed by the Schrödinger equation

$$\hat{H}\,|\Psi(t)\rangle = i\frac{\partial}{\partial t}\,|\Psi(t)\rangle, \tag{1}$$

with the total Hamiltonian, composed of the atomic binding potential ($\hat{V}$) and laser–electron interaction ($\hat{V}_{le}$), given by

$$\hat{H} = \frac{\hat{p}^2}{2} + \hat{V}_{le}(\boldsymbol{r}, t) + \hat{V}(\boldsymbol{r}). \tag{2}$$

The ionization amplitude, which describes the transition from an initial bound state $|\Psi_0(t)\rangle$ with ionization potential $I_p$ to a final continuum state $|\Psi_{\boldsymbol{p}}(t)\rangle$, defined by the Hamiltonian $\hat{H}$, is

$$T_{\boldsymbol{p}} = \lim_{t\to\infty, t'\to-\infty} \langle \Psi_{\boldsymbol{p}}(t)|\hat{U}(t,t')|\Psi_0(t')\rangle, \tag{3}$$

The total Hamiltonian's time-evolution operator fulfills the Dyson equation. If we incorporate the Dyson equation into the above equation, it can further be simplified. The initial term originating from $\hat{U}(t,t')$ gets annulled because the initial and final states are orthogonal, resulting in the remaining expression

$$T_{\boldsymbol{p}}^{(0)} = (-i)\lim_{t\to\infty, t'\to-\infty}\int_{t'}^{t} d\tau \langle \Psi_{\boldsymbol{p}}(t)|\hat{U}_{le}(t,\tau)\hat{V}_{le}(\boldsymbol{r},\tau)|\Psi_0(\tau)\rangle. \tag{4}$$

Here, we considered only "direct" electrons in the above expression. These are the electrons that, following the initial ionization, do not experience the binding potential anymore. To obtain the transition amplitude for these direct electrons, we substitute Volkov state $\langle \Psi_{\boldsymbol{p}}(t)|\,\hat{U}_{le}(t,\tau) \approx \langle \chi_{\boldsymbol{p}}(\tau)|$ which is given below by Equation (7). This substitution leads to the well-known SFA amplitude for direct electrons

$$T_{\boldsymbol{p}}^{(0)} = (-i)\int_{-\infty}^{\infty} d\tau \langle \chi_{\boldsymbol{p}}(\tau)|\hat{V}_{le}(\boldsymbol{r},\tau)|\Psi_0(\tau)\rangle. \tag{5}$$

By utilizing a hydrogen-like 1 s wave-function, it is possible to readily substitute the initial bound state with a modified ionization potential $I_p$ as

$$|\Psi_0(t)\rangle = |\Psi_0\rangle \, e^{iI_p t} = \frac{2I_p^{\frac{3}{2}}}{\sqrt{\pi}} e^{-\sqrt{2I_p}r} e^{iI_p t}. \tag{6}$$

The calculation of the matrix element (5) can be enlightening, particularly for situations with high intensity. To begin, we need to consider the explicit expression of the Volkov wave function, which in dipole case is given by

$$\chi_{\boldsymbol{p}}(\boldsymbol{r},\tau) = \frac{e^{-iS_v(\tau)}}{(2\pi)^{3/2}} e^{i\boldsymbol{p}\cdot\boldsymbol{r}}, \tag{7}$$

with the Volkov phase, $S_v(\tau)$ representing the classical action, given by

$$S_v(\tau) = \frac{1}{2} \int^\tau dt' [\boldsymbol{p} + \boldsymbol{A}(t')]^2. \tag{8}$$

The Volkov phase is a phase factor that appears in the wave function of a free electron in the presence of a strong electromagnetic field, which is described by the Volkov wave function (7). It represents the classical action of a free electron that occurs when the electron interacts with the laser field, represented by the vector potential $\boldsymbol{A}(t')$, and also depends on the electron momentum $\boldsymbol{p}$.

### 2.2. Vector Potential

To gain a more accurate understanding of the above-threshold ionization process, it is necessary to first describe the laser field using the vector potential, which incorporates both the spatial and temporal variations of the field. We begin with a vector potential that takes the form

$$\begin{aligned} \boldsymbol{A}(\boldsymbol{r},t) &= \int d^3\boldsymbol{k}\, \boldsymbol{A}(\boldsymbol{k},t), \\ \boldsymbol{A}(\boldsymbol{k},t) &= Re\{\boldsymbol{a}(\boldsymbol{k})e^{i(\boldsymbol{k}\cdot\boldsymbol{r}-\omega_k t)}\}, \end{aligned} \tag{9}$$

given by the arbitrary integral superpositions of plane-wave modes. Here, $\boldsymbol{a}(\boldsymbol{k})$ represents the complex Fourier coefficient.

In order to write the vector potential of a laser pulse in the form of Equation (9), we take into account an elliptically polarized laser pulse with an ellipticity of $\epsilon$ and helicity $\Lambda$. In addition, the overall pulse duration is equivalent to a whole number of optical cycles ($n_p$), such that $\tau_p = n_p T$, where $T = 2\pi/\omega$, and $\phi_{CEP}$ is the phase between the carrier wave and its envelope. Then the vector potential of a laser pulse can be expressed in the following manner:

$$\boldsymbol{A}(\boldsymbol{r},t) = \frac{A_0}{\sqrt{1+\epsilon^2}} f(\boldsymbol{r},t) \left( \cos(\mathrm{u}+\phi_{\mathrm{cep}})\boldsymbol{e}_x + \epsilon\Lambda\sin(\mathrm{u}+\phi_{\mathrm{cep}})\boldsymbol{e}_y \right). \tag{10}$$

Here, we used the short notation for $u = \boldsymbol{k}\cdot\boldsymbol{r} - \omega_0 t$. The function $f(\boldsymbol{r},t)$ is the envelope of the pulse and in this work we will consider a sin-squared envelope given as

$$f(\boldsymbol{r},t) = \begin{cases} \sin^2(\frac{\mathrm{u}}{2\mathrm{n_p}}), & 0 \le t \le \tau_p \\ 0, & \text{otherwise.} \end{cases} \tag{11}$$

Further, by expanding the trigonometric products by including the Equation (11) in Equation (10), we can modify the vector potential as

$$\boldsymbol{A}(\boldsymbol{r},t) = \sum_{j=-1}^{1} \frac{A_j}{\sqrt{1+\epsilon^2}} \left( \cos(\mathrm{u}_j+\phi_{\mathrm{cep}})\boldsymbol{e}_x + \epsilon\Lambda\sin(\mathrm{u}_j+\phi_{\mathrm{cep}})\boldsymbol{e}_y \right). \tag{12}$$

To this end, we expressed the vector potential of a laser pulse as the superposition of three monochromatic plane-wave beams each characterized by a different frequency labeled by $j$. Specifically, for $j = -1$, the frequency $\omega$ is given by $\omega = (1 - 1/n_p)\omega_0$. When $j = 0$, the frequency $\omega$ is equal to $\omega_0$, while for $j = 1$, the frequency is $\omega = (1 + 1/n_p)\omega_0$. The term $(k_j \cdot r - \omega_j t)$ is expressed as $u_j$. We decompose the $\sin^2$ envelope by expanding the trigonometric products. The resulting vector potential, which is given by Equation (12), is then plotted in three dimensions. The corresponding plot is shown in Figure 2, which provides a visual representation of the vector potential corresponding to two different pulse cycle ($n_p$).

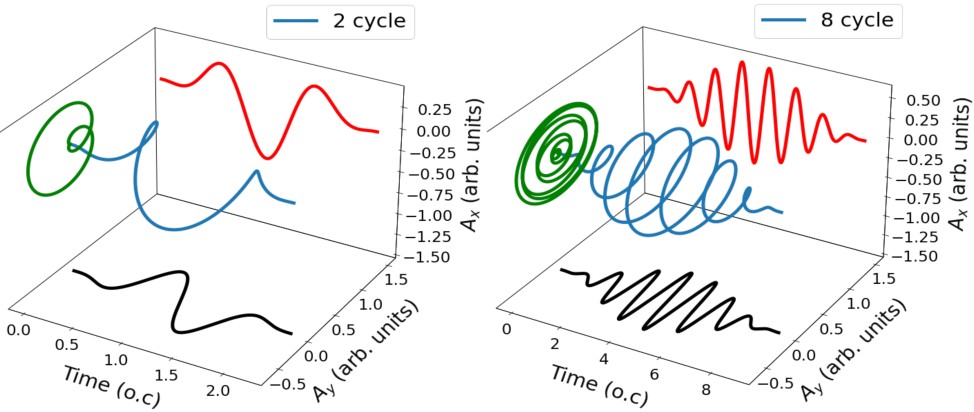

**Figure 2.** This figure depicts the vector potential Equation (12) of a right circularly polarized laser pulse (represented by the blue solid line) with a $\pi$-CEP, along with its projection onto the x-y plane (represented by the red, black, and green solid lines). The laser wavelength and intensity are maintained at 700 nm and $5 \times 10^{14}$ W/cm², respectively. The figure displays two different pulse durations: 2 cycles (**left**) and 8 cycles (**right**).

### 2.3. Nondipole Volkov State

The nondipole effects, caused by the magnetic part of the laser field, play a significant role in the dynamics of ionizing electrons. This magnetic component induces a transfer of momentum to the electrons, which can have a substantial impact on the ionization process. In the Volkov state, the laser field is incorporated through the Volkov phase, as indicated by Equation (8). However, to accurately account for the nondipole effects, the volkov phase must be modified. This modification must be carefully designed to accurately reflect the influence of the magnetic field in the ionization process. Therefore, incorporating the effects of the magnetic field and modifying the Volkov phase are essential steps in accurately modeling the dynamics of ionizing electrons in the presence of laser fields. In recent years, advancements in modifying the Volkov phase have enabled the observation of nondipole-induced effects in ATI. Notably, in Ref. [28], the Volkov state was derived for an infinitely extended laser beam, as represented by

$$\chi_p(r, t) = \frac{1}{(2\pi)^{\frac{3}{2}}} e^{-i(E_p t - p \cdot r)} e^{-i\Gamma(r,t)}. \tag{13}$$

The modified Volkov state presented in (13) has proven to be a valuable tool for investigating the effects of magnetic fields in ATI. The additional term $\Gamma(r, t)$, as the Volkov phase, accounts for the influence of the magnetic field on the electron's momentum. This modification allows for a more accurate description of the ionization process, particularly in the presence of strong electromagnetic fields. Despite the advancements made in modifying the Volkov phase, a limitation of this theory remained as it was initially restricted to continuous laser beams. However, through the description of the vector potential for a pulse as a superposition of plane wave beams, as shown in (12), similar formalism can be employed as demonstrated in our previous work [29]. The decomposition of the vector

potential in the form of (12) enables it to be written in the form of (9). The decomposition of the vector potential, in the form of Equation (12), provides a means for investigating the impact of nondipole effects in pulsed laser fields, which is described by the Volkov state

$$\chi_{\boldsymbol{p}}(\boldsymbol{r},t) = (2\pi)^{-3/2} \prod_{j=-1}^{1} \left[ \prod_{i=1}^{4} \sum_{n_i^j=-\infty}^{\infty} J_{n_i^j}(x_i^j) \times e^{-i(E_N t - \boldsymbol{p}_N \cdot \boldsymbol{r} - \Phi_N)} \right]. \tag{14}$$

The above equation, which is a modification of Equation (13), is a solution involving the vector potential described in Equation (12). The product over $j$ in this equation represents a superposition of individual frequencies of the vector potential (12). These frequencies are characterized by the Bessel function $J_{n_i^j}(x_i^j)$. In addition, the modified photoelectron energy ($E_N$), momentum ($\boldsymbol{p}_N$), and phase ($\Phi_N$) are given as

$$E_N = E_{\boldsymbol{p}} + N^{(0)}\omega_0, \tag{15a}$$

$$\boldsymbol{p}_N = \boldsymbol{p} + N^{(0)}\boldsymbol{k}_0, \tag{15b}$$

$$\Phi_N = (N^{(1)} + 2N^{(2)})\phi_{\text{cep}} + N^{(1)}\Lambda\varphi_p^{(\epsilon)}, \tag{15c}$$

respectively. Here, we introduced the short notation $N^{(0,1,2)}$, which depends on the pulse cycle and the $n_i^j$, which occur in Bessel functions and are given as

$$
\begin{aligned}
N^{(0)} &= \frac{3\alpha}{8} + (n_0^1 + 2n_0^2 + n_0^3 + n_{-1}^3 + n_0^4 + n_{-1}^4)C_0 \\
&\quad + (n_{-1}^1 + 2n_{-1}^2 + n_0^3 + n_1^3 - n_0^4 + n_1^4)C_{-1} \\
&\quad + (n_1^1 + 2n_1^2 + n_{-1}^3 + n_1^3 - n_{-1}^4 - n_1^4)C_1, \\
N^{(1)} &= n_0^1 + n_{-1}^1 + n_1^1, \\
N^{(2)} &= n_0^2 + n_{-1}^2 + n_1^2 + n_0^3 + n_{-1}^3 + n_1^3.
\end{aligned}
\tag{16}
$$

The arguments of the Bessel function (14), given in Table 1, include the modified pondermotive energy ($\alpha$) and the product of kinetic and field-induced photoelectron ($\rho_\epsilon$) momentum.

**Table 1.** Arguments ($x_i^j$) of the Bessel functions in Equations (14) and (18). The indices $i$ and $j$ are counted in the columns and rows, respectively.

| $x_i^j$ | 1 | 2 | 3 | 4 |
|---|---|---|---|---|
| $-1$ | $\rho_\epsilon \frac{D_{-1}}{C_{-1}}$ | $\alpha\frac{1-\epsilon^2}{1+\epsilon^2}\frac{D_{-1}^2}{2C_{-1}}$ | $2\alpha\frac{1-\epsilon^2}{1+\epsilon^2}\frac{D_0 D_1}{C_0+C_1}$ | $2\alpha\frac{D_0 D_1}{C_0-C_1}$ |
| $0$ | $\rho_\epsilon \frac{D_0}{C_0}$ | $\alpha\frac{1-\epsilon^2}{1+\epsilon^2}\frac{D_0^2}{2C_0}$ | $2\alpha\frac{1-\epsilon^2}{1+\epsilon^2}\frac{D_0 D_{-1}}{C_0+C_{-1}}$ | $2\alpha\frac{D_0 D_{-1}}{C_0-C_{-1}}$ |
| $1$ | $\rho_\epsilon \frac{D_1}{C_1}$ | $\alpha\frac{1-\epsilon^2}{1+\epsilon^2}\frac{D_1^2}{2C_1}$ | $2\alpha\frac{1-\epsilon^2}{1+\epsilon^2}\frac{D_{-1} D_1}{C_{-1}+C_1}$ | $2\alpha\frac{D_{-1} D_1}{C_{-1}-C_1}$ |

Specifically, the constants C's and D's are given by

$$
\begin{aligned}
C_0 = 1, \quad C_{-1} = 1 - \frac{1}{n_{\text{p}}}, \quad C_1 = 1 + \frac{1}{n_{\text{p}}}, \\
D_0 = \frac{1}{2} \quad \text{and} \quad D_{-1} = D_1 = \frac{-D_0}{2}.
\end{aligned}
\tag{17}
$$

### 2.4. Transition Amplitude and Ionization Probability

The process of ATI is governed by the transition amplitude (5), which describes the probability of an electron undergoing a transition from an initial bound state to a final continuum state. Having obtained the nondipole Volkov states in their final form (14), we

can now proceed to calculate the direct SFA transition amplitude for the ATI of atomic targets and compute the corresponding photoionization probability. By substituting the nondipole Volkov states Equation (14) and the initial state (6) in (4), we can obtain the required expression as

$$T_{\boldsymbol{p}}^{(0)} = \prod_{j=-1}^{1} \left[ \prod_{i=1}^{4} \sum_{n_i^j=-\infty}^{\infty} J_{n_i^j}(x_i^j) \times \left[ \frac{\langle \boldsymbol{p}_N | V(\boldsymbol{r}) | \Psi_0 \rangle}{I_p + E_N} + \langle \boldsymbol{p}_N | \Psi_0 \rangle \right] \times \left[ 1 - e^{i(I_p + E_N)\tau_p} \right] \right], \quad (18)$$

To determine the photoionization probability P(p), we express it in terms of the SFA transition amplitude as

$$P(\boldsymbol{p}) = \frac{|T_{\boldsymbol{p}}|^2 d^3 \boldsymbol{p}}{d\Omega_{\boldsymbol{p}} dE_{\boldsymbol{p}}} = p|T_{\boldsymbol{p}}|^2 \approx p|T_{\boldsymbol{p}}^{(0)}|^2, \quad (19)$$

describing the probability for an electron emitted with energy $E_{\boldsymbol{p}} = \frac{p^2}{2}$ into the solid-angle $d\Omega_{\boldsymbol{p}}$. In this context, we approximate the full transition amplitude $T_{\boldsymbol{p}}$ by the direct amplitude $T_{\boldsymbol{p}}^{(0)}$, which is defined by Equation (18).

## 3. Results and Discussion

In the preceding sections, we have introduced an explicit formulation for the ATI transition amplitude under the nondipole SFA. In this section, we will deliberate on the outcomes that can be achieved for specific laser parameters. The impact of the Lorentz force on the strong-field ionization of atoms was initially observed in the momentum distribution of emitted photoelectrons. While photoelectrons typically follow the electric field vector perpendicular to the laser propagation, slight shifts in electron momenta have been observed in the direction parallel or antiparallel to the laser propagation, depending on the gas target, intensity, wavelength, and pulse cycles of the laser field. Specifically, we will limit our discussion to circularly polarized beams. Our focus will be on the dependence of peak shift in the ATI spectra on the pulse cycle, intensity, and atomic target.

As the photoelectron's longitudinal motion in the laser field escalates with the laser wavelength $\lambda$ and intensity $I$, the influence of the nondipole effects on the ATI can be regulated by these parameters. In particular, peak offsets can be gauged in near-IR at intensities ranging from $10^{14}$ W/cm$^2$ to $10^{15}$ W/cm$^2$. In our analysis, emphasis is placed on the peak shift of the ATI spectra. To determine this parameter, we perform calculations for the ATI spectra (19) at different polar angle values ranging from $\theta_p = 0$ to $\pi$. Subsequently, we extract the energy ($E_{\boldsymbol{p},max}$) and the corresponding polar angle ($\theta_{p,max}$), where the ionization probability reaches its maximum value. By utilizing this information, we can derive the peak shift value, denoted as $\Delta p_z = \sqrt{2E_{\boldsymbol{p},max}\theta_{p,max}}$. Several strong-field ionization experiments have identified a nonzero $\Delta p_z$ component of photoelectron momenta along the laser propagation direction in near to mid-IR driving fields. This phenomenon, known as peak shift $\Delta p_z$ in current literature, was observed in experiments [24–27]. Velocity map imaging was used by Smeenk et al. [23] to measure momentum distribution in the x-z plane for circularly polarized driving laser fields of varying intensity and wavelength at 800 and 1400 nm, yielding pioneering results. The measurements at these wavelengths found peak shifts $\Delta p_z$ ranging from 5 to $20 \times 10^{-3}$ a.u., corresponding to roughly 5 to 10 photon momenta. Additionally, these shifts were observed to increase linearly with laser intensity. By analyzing the changes in momentum of photoelectrons at various polar angles, one can gain a more comprehensive understanding of the effects of the electromagnetic field on electron's behavior.

In the context of ATI, pulse cycles refer to the number of optical cycles of the laser pulse during the ionization process. In nondipole ATI, ionization occurs under the influence of a strong laser pulse, where the electron is ionized from the atom or molecule and is accelerated in the presence of the field. The peak shifts in the non-dipole ATI are caused

by asymmetry in the electron wave packet created during the ionization process. This asymmetry arises because of the influence of the magnetic field component of the laser pulse, which breaks the symmetry of the dipole approximation. Increasing the number of pulse cycles in nondipole ATI has a pronounced effect on peak shifts, leading to a larger displacement of the ionization peaks from their positions in the nondipole approximation. This displacement is caused because the electron-wave packet has more time to evolve and becomes more asymmetric with each additional pulse cycle. Additionally, increasing the number of pulse cycles also lead to changes in the overall shape of the energy spectra, including the appearance of additional peaks or the disappearance of existing peaks [30]. These changes occur because of the interference between the electron wave packets created in each pulse cycle. Clearly, our focus here is to only see the effect due to optical cycles of a pulse. Figure 3 displays such effects in the peak shifts that occur in the momentum of a photoelectron in the propagation direction. The interaction between the released electron and the laser pulse's electric field strongly depends on the temporal structure of the pulse. Short pulses with few cycles have a more pronounced effect on the electron dynamics, leading to significant peak shifts.

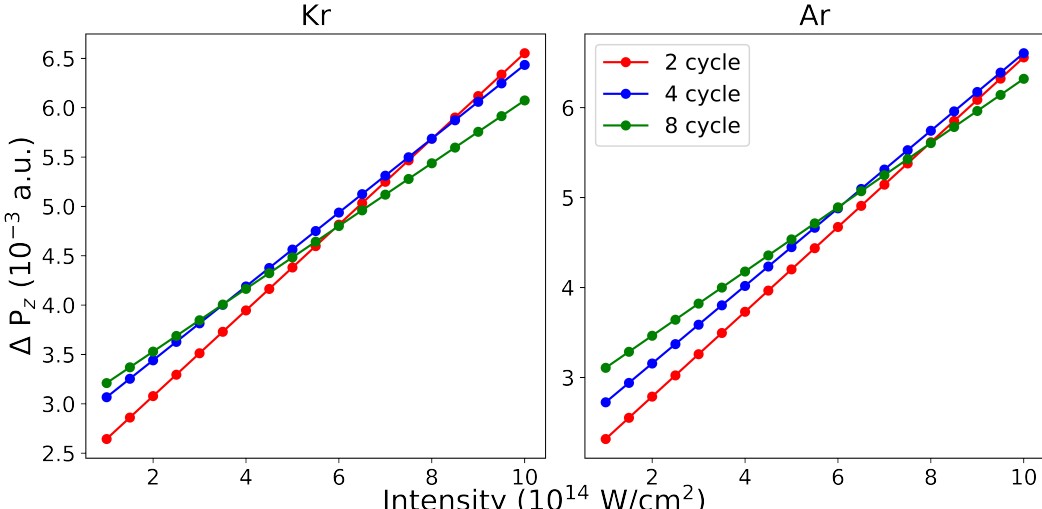

**Figure 3.** The peak shift $\Delta P_z$ of the maxima in ATI spectra are plotted as a function of laser intensity I. The laser parameters used are described in Figure 2. Results are shown for two different atomic targets, Kr (**left**) and Ar (**right**).

To investigate the influence of the optical cycles on the ionization probability in ATI, we varied the number of optical cycles in the laser pulse and calculated the ionization probability of the emitted photoelectrons for each case corresponding to the polar angle at which the peak shift occurs. Figure 4 shows that the maximum ionization probability strongly depends on the number of optical cycles of the laser pulse. As the number of cycles increased, the maximum ionization probability also increased. For example, for a laser pulse with a wavelength of 700 nm and a peak intensity of $8 \times 10^{14}$ W/cm$^2$, the maximum ionization probability increases from approximately 25% for a 2-cycle pulse to over 80% for an 8-cycle pulse. In addition to the number of cycles, the ionization probability is sensitive to the intensity of the laser field in the range of $1 \times 10^{14}$ to $1 \times 10^{15}$ W/cm$^2$. Higher laser intensities led to a higher probability of ionization, as expected from the strong-field ionization mechanism. For instance, for a four-cycle pulse, the maximum ionization probability increases from approximately 45% at an intensity of $5 \times 10^{14}$ W/cm$^2$ to over 80% at an intensity of $1 \times 10^{15}$ W/cm$^2$. These results suggest that the ionization probability can be further enhanced by increasing the laser field intensity within this range. At lower intensities, as seen in the sub-figure within Figure 4 on a logarithmic scale, the ionization probability is lower for higher pulse cycles because the ionization mechanism in this regime is dominated by multiphoton ionization (MPI). The so-called Keldysh pa-

rameter ($\gamma$) characterizes the different strong-field regimes. For a given atomic target interacting a specific laser pulse, the Keldysh parameter is calculated by $\gamma = \sqrt{I_p/2U_p}$. This dimensionless parameter only depends on the ionization potential ($I_p$) of the target atom and the pondermotive energy ($U_p$). From Figure 4 one can see how the Keldysh parameter varies over intensity. For $\gamma > 1$, the MPI and ATI dominates the strong-field regime while for $\gamma < 1$ tunnel ionization dominates. In tunneling ionization, the electron tunnels through the potential barrier created by the laser field are ionized. The probability of tunneling depends on the interaction time between the laser field and electron, which is proportional to the number of pulse cycles. However, when the laser intensity is low, the electric field strength is not strong enough to efficiently ionize the atom, and the ionization probability is low. At such intensities, increasing the number of pulse cycles may not be sufficient to overcome the ionization barrier and the ionization probability remains low.

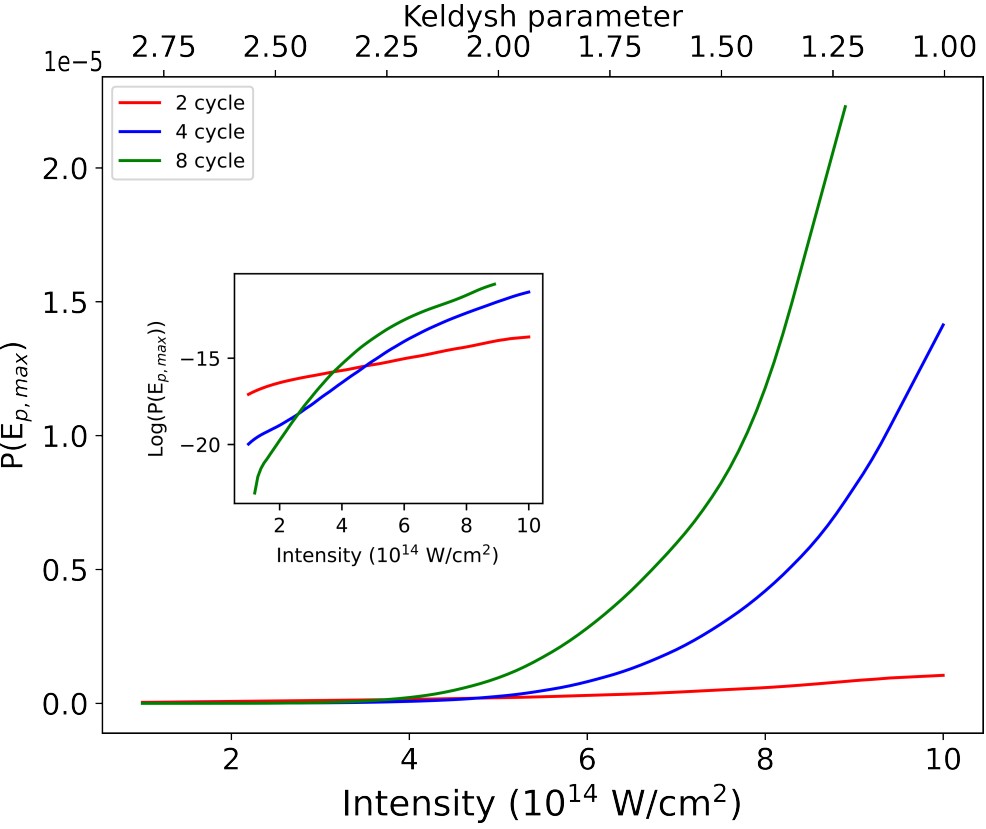

**Figure 4.** This figure depicts the ionization probability as a function of intensity for different pulse cycles. The subplot is plotted by taking the log of ionization probability. The laser parameters are the same as in Figure 2. The atomic target considered is Krypton with $I_p$ = 13.9996 eV.

## 4. Conclusions

We utilized the nondipole approach of the SFA as a tool to examine the ATI phenomenon in noble gas atoms. This analysis encompassed a circularly polarized near-IR plane-wave laser pulse. Within this context, the nondipole Volkov states employed in our study align with the expressions already established in earlier research. Specifically, we focused on scrutinizing the peak shift (denoted as $\Delta p_z$) of the maxima present in the ATI spectra in relation to the laser's propagation direction.

We investigated the effect of the laser parameters, namely intensity and pulse cycles, on the peak shifts observed in the momentum distribution of photoelectrons in the direction of laser propagation under nondipole ATI. Our investigation revealed that the longitudinal motion of the photoelectrons in the laser field is affected by the laser intensity as well as the optical cycles and laser carries. Specifically, we identified a peak shift in the ATI spectra, which could be regulated by varying the laser pulse cycles and intensity. Our results show



that the peak shifts increase with the laser intensity and are more pronounced in shorter pulses. We also demonstrate that the maximum ionization probability strongly depends on the number of optical cycles of the laser pulse. These findings contribute to a more comprehensive understanding of the effects of electromagnetic fields on the behavior of electrons and provide valuable insights for future strong-field ionization studies.

**Author Contributions:** Methodology, D.F.D.; writing—review and editing, D.F.D. and S.F. All authors have read and agreed to the published version of the manuscript.

**Funding:** This work has been funded by the Deutsche Forschungsgemeinschaft (DFG, German Research Foundation)—440556973 and also by the Research School of Advanced Photon Science (RS-APS) of Helmholtz Institute Jena, Germany.

**Institutional Review Board Statement:** Not applicable.

**Informed Consent Statement:** Not applicable.

**Data Availability Statement:** Not applicable.

**Conflicts of Interest:** The authors declare no conflict of interest.

**Abbreviations**

The following abbreviations are used in this manuscript:

SFA Strong Field Approximation
ATI Above Threshold Ionization

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
