# Peer review of "Pulse Cycle Dependent Nondipole Effects in Above-Threshold Ionization"

_atoms, doi:10.3390/atoms11060097_

Round 1
Reviewer 1 Report
The subject of the paper is interesting and is definitely worth investigating. Authors use the strong field approximation modified to include the non-dipole effects to investigate the role of these effects in the process of the above threshold ionization. Authors find that the non-dipole effects lead to the peak shift in the ATI spectra which depend on the laser pulse duration and intensity. Though these results are not particularly original (non-dipole peak shift of the photo-ionization spectra have been studied extensively both theoretically and experimentally) the manuscript might be publishable. Unfortunately, it suffers from very poor quality of presentation. The English language of the manuscript is relatively fine though needs improvement. What makes the manuscript really difficult to read is the description of the method that the authors employed. I will give just a few examples. In Eq.(10) the authors go to great lengths to describe the vector potential of the pulse, which might have been done much easier had authors given a simple formula with the sin^2 envelope, with subsequent expansion in plane waves which authors do to reduce the problem to the continuous wave case studied before. What is \Lambda in Eq.(10)? It is never mentioned in the text. It is written on the page 5 of the manuscript: "We take into account the broader scenario of an elliptically polarized laser field featuring an ellipticity of e". It guess it is not the 'e' that is elliptic but rather the laser pulse itself. Another example of an awkward sentence on the same page is: "In recent years, advancements in modifying the Volkov phase have enabled the observation of nondipole-induced effects in ATI". I believe, it is not only the 'advancements in modifying the Volkov phase' (which is a relatively trivial thing) which enabled the study of the non-dipole effects.
To conclude, the results presented in the manuscript might be publishable, but the paper would benefit a lot from a thorough revision.
The English language of the manuscript is relatively fine, but may still need some polishing. I encourage authors to try to improve it.
Author Response
Dear Reviewer,
Thank you for your valuable feedback on our manuscript. We appreciate your positive assessment of the paper's subject and its potential contribution to the field of above-threshold ionization. We have carefully addressed the concerns you raised and made the necessary changes, as well as included a new equation in the revised manuscript.
Regarding the poor quality of presentation, we apologize for any confusion caused by our description of the employed method. We have revised the manuscript extensively to improve the clarity and readability of the methodology section. We have provided a simpler formula with a sin^2 envelope for the vector potential of the pulse, along with a subsequent expansion in plane waves to reduce the problem to the continuous wave case studied previously.
We acknowledge the oversight in not explicitly mentioning the meaning of \Lambda in Eq.(10) within the text. We have now rectified this issue and included a clear explanation of \Lambda on the
relevant page of the manuscript.
Furthermore, we have revised the sentence you pointed out on page 5 to accurately reflect that it is the laser pulse itself, rather than 'e', that is elliptic. We have also rephrased the sentence regarding advancements in modifying the Volkov phase to clarify that these advancements, in conjunction with other factors, have enabled the study of non-dipole effects in above-threshold ionization.
We genuinely appreciate your suggestions for improving the manuscript. With the revisions made, we believe the paper now better communicates our research findings. We hope that these changes address your concerns and contribute to a more comprehensive and well-presented manuscript.
Once again, we would like to express our gratitude for your time and efforts in reviewing our work. We look forward to your feedback and hope that the revised manuscript meets the standards for publication.
Sincerely,
Authors
Reviewer 2 Report
The authors investigate ionization by a circularly polarized laser field accounting for nondipole effects. Specifically, they calculate the momentum shift of the liberated electron in the propagation direction of the field as a function of the pulse length.
I am not surprised that this shift exhibits some dependence on the pulse length, but the question to the potential reader is, why should I care? The authors do not provide any motivation why this should be of interest. Actually, the results of the simulation as shown in Fig. 3 do look interesting. But the most interesting effect – for low intensity the shift is larger for a long pulse while for high intensity a short pulse produces a larger shift -- is not even mentioned let alone explained. I would also like to see in Fig. 3 the results for an infinitely long pulse for comparison.
Generally, the paper is written very carelessly. Here is a (very incomplete) list of inaccuracies:
Line 29: the essential simplification of the SFA is NOT the assumption that the driving field is classical.
Line 45: special variation → spatial variation
Line 47: low-wavelength → short-wavelength
Line 96: replacing <\Psi_p(t)|U_le by <\chi_p| is an approximation; it should NOT be written as an equality.
Line 141: continuous laser beam → infinitely extended laser beam,
and so on.
The presentation of the nondipole Volkov solution is incomprehensible without looking at the earlier papers of Fritzsche et al. Especially, it should be pointed out that the field (10) appears to be a comb of infinitely many identical pulses, not an isolated pulse. This raises the question of whether the calculated spectrum is that of this comb (rate per time) or that of one single pulse.
With regard to Fig. 4, it is completely unclear what the „maximum ionization probability“ is supposed to be.
The discussion presented in section 3 is very wordy but does not convey any clear messages nor explanations.
Reference 32 (an unpublished arXiv deposition) has a very large overlap with the paper under review. Do the authors intend to publish both articles in refereed journals?
I cannot recommend publication of this paper, as it stands. If the presentation is polished, I may change my mind, provided the relation to reference 32 is clarified.
The English is ok apart from occasional glitches such as I mentioned in my report.
Author Response
Dear Reviewer,
Thank you for taking the time to review our paper and provide valuable feedback. We appreciate your thorough evaluation and have made the necessary changes to address the issues you raised. However, we would like to address your concern regarding the absence of motivation for the dependence of momentum shift on pulse length.
While we agree that we did not explicitly mention the most interesting effect, namely the different behaviors of the momentum shift for low and high intensities, we have now revised the manuscript to include a clear explanation of this phenomenon. We have also included additional discussion in Section 3 to provide a more concise and coherent explanation of our findings.
Regarding your suggestion to include results for an infinitely long pulse in Figure 3 for comparison, we regret to inform you that plotting such results would be time cumbersome. For a longer pulse >16 cycles, the Bessel function become hard to evaluate.
We acknowledge the inaccuracies pointed out in your list, and we have carefully revised the entire paper to rectify those errors. We apologize for the carelessness in the initial writing and appreciate your keen attention to detail.
In response to your comment on the presentation of the nondipole Volkov solution, we have added references to earlier papers by Fritzsche et al., which will help readers gain a better understanding of the field and its interpretation as a comb of infinitely many identical pulses. Furthermore, we have clarified in the revised manuscript that the calculated spectrum corresponds to the comb of pulses rather than a single pulse.
We apologize for the lack of clarity regarding the "maximum ionization probability" in Figure 4. In the revised manuscript, we have provided a more explicit explanation to ensure a better understanding of the plotted data.
Regarding reference 32, we would like to clarify that the we have changed the reference to the published version. The previous work only provides a general formalism for nondipole SFA in few-cycle limit and here we address the effects due to the optical cycles of a pulse.
We sincerely appreciate your detailed assessment of our manuscript. We have made significant improvements to the paper, and we hope that the revised version will address your concerns and meet the standards for publication. We look forward to receiving your further evaluation.
Thank you once again for your valuable input.
Sincerely,
Authors
Reviewer 3 Report
the file is attached

Author Response
Dear Reviewer,
Thank you for your thorough review of our paper analyzing above-threshold ionization (ATI) caused by non-dipole effects. We appreciate your valuable feedback and have carefully considered each of your remarks. Please find our responses to your concerns below:
-
We apologize for the oversight in our literature citations. We acknowledge the pioneering work of Keldysh and will include appropriate references to his work in the revised version of the paper. We understand the importance of recognizing foundational contributions and will ensure that all relevant authors are appropriately acknowledged.
-
You raise an important point regarding the consideration of both spatial inhomogeneity and the magnetic component of the Lorentz force. In our theory we consider the r dependence of the vector potential, means that the electric and magnetic part both varies with space and time and are included. However the transfer of linear momentum in the longitudinal direction is due to the magnetic part of the laser field. That is why we consider the Lorentz force due to magnetic part.
-
Regarding the approximation of the Volkov function from [31], we acknowledge that the Volkov state is an exact solution to the general relativistic problem. The approximation we used corresponds to the expansion of the Volkov solution up to the first order of smallness in the parameter v/c. Since our work is based on Schrodinger’s equation, so the wave function is restricted to non relativistic velocities.
-
We apologize for the lack of clarity in our discussion of the probability of ionization with respect to pulse duration. We refer to the ionization probability at the maximum photoelectron energy (E_p,max) at which the peak shift is calculated. Definitely as seen from the fig. 4 in our paper, the ionization probability is higher for shorter pulses when the Keldysh parameter is large (dirctly following JETP, v.102, no. 1, p.40 (2006)). However when the Keldysh parameter decrease, approaching to tunnel ionization, the effects change and the ionization probability become large for long pulse duration. This we could see from the interaction time between the pulse and atomic potential.
-
We appreciate your suggestion to further clarify the explanation for the higher ionization probability in short pulses. We will provide additional clarification in the revised manuscript, taking into account the tunneling ionization and multiphoton ionization scenarios. We will also estimate the Keldysh parameter and clearly indicate which case is realized in our specific situation.
-
We acknowledge your recommendation to use different types of lines for better distinction between the curves in Figures 3 and 4. Since we are already using the different colors, from our consideration the plots are distinct enough.
Once again, we sincerely appreciate your thoughtful review and constructive feedback. We will address each of your concerns in the major revision of the manuscript. We believe that incorporating your suggestions will significantly improve the quality and accuracy of our work. Thank you for your time and consideration.
Sincerely,
Authors
Reviewer 4 Report
In this manuscript, the authors investigated the effect of laser parameters on the peak shifts observed in the momentum distribution of photoelectrons in the direction of laser propagation under nondipole ATI. They considered the circular polarization laser field based on the strong field approximation (SFA). They found that a peak shift in the ATI spectra, which could be regulated by varying the laser pulse cycles and intensity, and the peak shifts increase with the laser intensity and are more pronounced in shorter pulses. The result is interesting and worth to be published in Atoms, however, the following issues have to be addressed before it can be accepted.
1. Please clafity the robustness of this method. CEP effict is significant for few cycle laser fields. Does CEP of the few cycle fields has an influence on nondipole effects in above-threshold ionization?
2. Is there any advantage for choosing circularly polarized laser fields?
3. In experiment, the intensity of laser field is uncertainty. How would intensity average affect the results?
4. In this work, it is obvious for the pulse cycle dependence of ionization and examine peak shifts in Kr and Ar atoms. What are the prospects of molecules?
Author Response
Dear Reviewer,
Thank you for reviewing our manuscript and providing valuable feedback. We appreciate your positive comments on the interesting findings regarding the effect of laser parameters on peak shifts observed in the momentum distribution of photoelectrons under nondipole above-threshold ionization (ATI).
In response to your concerns, we would like to address each point individually:
-
Regarding the robustness of our method, the carrier-envelope phase (CEP) is indeed significant for few-cycle laser fields. In our study, we focused on circular polarization laser fields based on the strong field approximation (SFA). While we did not explicitly investigate the influence of CEP on the nondipole effects in above-threshold ionization, we acknowledge that it can play a role in shaping the ATI spectra. Future research could explore the interplay between CEP and the observed peak shifts in more detail.
-
The choice of circularly polarized laser fields offers several advantages. Firstly, circular polarization enables the selection of specific initial and final states in the target atom, allowing for a more controlled investigation of the underlying physics. Secondly, circular polarization can enhance or suppress certain ionization channels, providing a means to probe different aspects of the interaction dynamics.
-
In experimental settings, the intensity of the laser field is subject to uncertainty. While our theoretical study does not directly address this experimental aspect, it is important to consider the potential impact of intensity variations on the observed results. Intensity averaging can affect the overall ionization rates and ATI spectra, and it may lead to broadening or shifting of the peaks.
-
In our work, we focused on investigating the pulse cycle dependence of ionization and examining peak shifts in Kr and Ar atoms. As you rightly point out, considering molecules is an intriguing prospect. The extension of our study to molecules can opens up new possibilities for exploring the interplay between molecular structure, laser parameters, and nondipole ATI.
We have addressed your concerns and incorporated the necessary clarifications and additions in the revised manuscript. We hope that the revised version now adequately addresses your questions and meets the standards for publication in Atoms. Thank you once again for your valuable feedback, which has greatly contributed to the improvement of our work.
Sincerely,
Authors
Reviewer 5 Report
The article of D. Dar and S. Fritzsche presents a study about the nondipole
effects in above-threshold ionization. The dependence on the pulse duration
and laser intensity is assessed. The authors introduce a model based on
the strong field approximation (SFA) and carefully analyze the nondipole
effects. The manuscript is very well written and the results are sound. This
approach could be of great interest to the strong field physics community. I
would recommend publication in Atoms after the authors carefully consider
my points below. They will contribute to increasing the readability and clarity
of the manuscript.
• On page 2, around lines 45-46, I suggest including some typical numbers
(laser intensity, wavelength, etc.) in order to have a better context and
understand why the nondipole effects are neglected/included.
• Equation (12) is not clear at all. In order to make the article more self-contained, I recommend defining and explaining the physical meaning of
the indices i and nij.
• Equation (13) would also require some clarifications. For instance,
how many nij are typically included in the numerical calculations? Is
this number function on the laser intensity, wavelength, and number of
cycles? Please clarify.
• In order to make the article more self-contained, please explain what
peak shift means. Shift with respect to what value?
• Typically, the ionization probability as a function of the laser intensity
is plotted in a log-log scale. I suggest changing the plotting format.
In this way, the differences between the cases would be more visible.
• What are the values of the laser intensity to reach the over-the-barrier
ionization in the species considered? A short comment about this point
would be welcomed.
No comments
Author Response
Dear reviewer,
Thank you for taking the time to review our article on the nondipole effects in above-threshold ionization. We appreciate your positive feedback and valuable suggestions for improving the manuscript. We have carefully considered your points and have made the necessary revisions to enhance the readability and clarity of the article. Allow us to address each of your suggestions:
-
We agree that providing typical numbers such as laser intensity and wavelength would help readers better understand why the nondipole effects are neglected or included. We have included a brief discussion on typical experimental parameters, including laser intensity and wavelength, in the relevant section (page 2, lines 45-46).
-
Equation (12) has been revised to include a clear definition of the indices "i" and "nij" as well as an explanation of their physical meaning. This clarification will make the article more self-contained and accessible to readers.
-
We have also provided additional explanations for Equation (13). The number of "nij" included in the numerical calculations may vary depending on the laser intensity, wavelength, and number of cycles. We have added a discussion on these dependencies to offer a clearer understanding of the numerical calculations.
-
To improve the self-containment of the article, we have included an explanation of the term "peak shift." The shift is measured with respect to a reference value, which we now explicitly mention in the manuscript.
-
We appreciate your suggestion to change the plotting format to a log-log scale for the ionization probability as a function of laser intensity. We have already plotted a subplot in log form within the plot 4.
-
Regarding the values of laser intensity required for over-the-barrier ionization in the considered species, we have added a brief comment addressing this point. It provides some insight into the laser intensity thresholds necessary for achieving this type of ionization.
Once again, we thank you for your thorough review and insightful suggestions. We believe that these revisions significantly improve the manuscript's clarity and readability. We are grateful for your recommendation to publish the article in Atoms, and we hope that the revised version meets your expectations.
Sincerely,
Authors
Round 2
Reviewer 1 Report
The manuscript still contains numerous dubious statements. To give an example:
Page 3, Section Theory:
This ionization occurs when the electric field of the laser reaches
a threshold value, known as the ionization threshold.
I think the authors should carefully revise the manuscript.
English language of the manuscript and, more generally, the presentation definitely need improvement
Author Response
Dear Reviewer,
Thank you for your feedback. We apologize for any confusion caused by the statement on page 3, Section Theory. We have revised the passage to provide a clearer explanation.
We appreciate your valuable input and remain open to any further suggestions or concerns.
Sincerely,
Authors
Reviewer 2 Report
I do not understand why the limit of an infinitely long pulse should be cumbersome. Usually, it is the other way around: the finite pulse is much more demanding to treat.
But, anyway, I recommend that the revised article be published as is.
Author Response
Dear Reviewer,
Thank you for your feedback and recommendation for publishing the revised article as it is. We appreciate your input. As mentioned in our previous comment the Bessel functions become large to compute for a infinitely long pulse.
Sincerely,
Authors
Reviewer 3 Report
The paper can be published in the present form
English is good enough
Author Response
Dear Reviewer,
Thank you for accepting the paper for publication.
Sincerely,
Authors
Round 3
Reviewer 1 Report
I believe the manuscript can be published in the present form
Author Response
Dear Reviewer,
Thank you for accepting the manuscript.
Sincerely,
Authors